# Visual Field Improvement by Standardized Automated Perimetry Following Panmacular Subthreshold Diode Micropulse Laser (SDM) in Open-Angle Glaucoma and Other Optic Atrophies [note 1]

**DOI:** 10.3390/diagnostics15070912

**Published:** 2025-04-02

**Authors:** Jeffrey K. Luttrull, Sathy V. Bhavan

**Affiliations:** Ventura County Retina Vitreous Medical Group, 3160 Telegraph Rd., Suite 230, Ventura, CA 93003, USA

**Keywords:** subthreshold laser, automated perimetry, glaucoma, optic neuropathy, neuroprotection, neuroenhancement

## Abstract

**Purpose:** To assess the effect of subthreshold diode micropulse laser (SDM) on visual fields (VF) by standardized automated perimetry (SAP) in open-angle glaucoma (OAG) and other non-glaucomatous optic atrophies (OA). **Methods:** The electronic medical records in a vitreoretinal practice were searched to identify the cohort of eyes with OAG and OA that underwent SAP before and after the initial SDM meeting study inclusion and exclusion criteria. Recorded data included mean deviation (MD), mean sensitivity (MS), and pattern standard deviation (PSD) before and after treatment. **Results:** A total of 55 eyes of 29 patients, 17 female, aged 62–89 years (avg. 76), with 48 eyes having OAG and 7 with OA, were included in the study. All SAP tests were performed the same day prior to the first SDM treatment, and the postop SAP within one month post-treatment. There were three groups: 36 total treated eyes, 14 treated simultaneously in both eyes prior to repeat SAP, and 22 treated in one eye prior to repeat SAP, along with 19 untreated fellow eye controls. Following SDM, MD and MS were significantly improved in all treated eyes and unilaterally treated eyes (*p* range 1.0 × 10^−4^ to 7.02 × 10^−6^). Untreated fellow eyes were also significantly improved (*p* = 0.03 for both MD and MS), but the MD and MS improvements in the treated eyes were significantly greater than untreated fellow eyes (*p* = 0.016 for both MD and MS). **Conclusions:** Panmacular SDM significantly improved VF by SAP in eyes with OAG and OA. This finding has important implications for management in both conditions.

## 1. Introduction

Chronic progressive retinopathies (CPRs) are the main causes of irreversible visual loss at all ages worldwide [1]. Panmacular low-intensity/high-density subthreshold micropulsed diode laser (SDM^TM^) has been shown to improve all measures of retinal and visual function in all CPRs examined, including age-related macular degeneration (AMD), diabetic retinopathy (DR), and inherited retinal degenerations (IRDs), without adverse treatment effects [2,3,4].

All CPRs are neurodegenerative conditions. OAG is also a neurodegenerative condition [2,3,4,5,6,7,8,9]. Like the responses to SDM treatment in other CPRs, significant improvements in visual acuity (VA), mesopic visual function, ganglion cell function by pattern electroretinography (PERG), and optic nerve function by visually evoked potential (VEP) have been reported following panmacular SDM treatment of the posterior retinal pigment epithelium (RPE) in eyes with OAG, suggesting the presence of an intrinsic retinopathy in OAG [2,3,4]. Recently, long-term treatment of eyes with OAG by regular periodic SDM maintenance therapy (“vision protection therapy”, VPT) has been found to inhibit optic glaucomatous atrophy by improving and restoring optical coherence tomography (OCT) measurements of retinal nerve fiber (NFL) and ganglion cell complex (GCC) thicknesses in VPT-treated eyes compared to eyes managed conventionally with intraocular pressure (IOP) control alone [10].

Studies of standardized automated perimetry (SAP) in OAG and other optic atrophies (OA) focus almost exclusively on detecting and characterizing visual field defects and documenting disease progression. Improvements in SAP in OAG and other optic atrophies (OA) are not seen in response to treatment except as testing artifact [11,12,13,14,15,16,17,18]. However, examples of improved VF by SAP in OAG, OA, and other CPRs have been described following SDM [3,8,9,10]. Despite these case reports, there has been no systematic study of the effects of SDM on VFs by SAP. In the current study, we retrospectively examine and statistically analyze the VF results by SAP in eyes with OAG and OA before and after initial treatment with SDM in an effort to better understand if SAP improvements following SDM in OAG and OAG are aberrations or represent a characteristic therapeutic response to treatment. Because visual field loss is a hallmark of disease progression in OAG, the ability to therapeutically reverse visual field loss would represent an important advancement in disease management and vision loss prevention in OAG.

## 2. Methods

This study examines the results of automated perimetry in eyes with OAG and OA treated with SDM in a vitreoretinal subspecialty practice consisting of a main clinic and two satellite clinics. This study was exempted from review by the Western Investigational Review Board as a retrospective study of patient-identified electronic medical record (EMR) data and complied with both the Health Insurance Portability and Accountability Act of 1996 and the tenets of the Declaration of Helsinki. As this study involves the use of only patient-identified EMR data and is thus exempt from IRB review, patient consent was neither required nor obtained. The datasets used and/or analyzed during the current study are available in Appendix A. The perimetry records and EMRs of all eyes that underwent SAP since the introduction of SAP into practice from 2020 to 2023 were reviewed to identify eyes with OAG or OA that underwent SAP before and after the initial SDM treatment.

The indication for SDM treatment in the eyes examined in this study was to slow disease progression, as predicted by prior studies showing improvements in prognostic surrogates of disease worsening following SDM, including all measures of retinal, optic nerve, retinal ganglion cell, and visual function in multiple disorders, including OAG [2,3,4,5,6,7,8,9,10].

Inclusion criteria were a prior diagnosis of OAG or OA. In OAG eyes, glaucomatous optic atrophy with visual field loss, and current and/or prior IOP-lowering treatment were required. This included current or prior use of topical medications, selective laser trabeculoplasty, and/or prior anterior segment pressure-lowering surgery. Inclusion also required sufficient mental acuity and manual dexterity to adequately perform SAP testing [11,12,13,14,15,16,17,18]. Exclusions included VA of 20/200 or worse, poor-quality/unreliable testing, prior SDM or other macular laser treatments, prior or current intraocular anti-vascular endothelial growth factor (VEGF) or steroid injections, and/or other obfuscating ocular disease such as significant cataracts, or macular diseases such retinal vascular occlusion, epiretinal membranes, macular edema, significant diabetic retinopathy (beyond “early” non-proliferative stage), prior retinal detachments, or advanced AMD.

### 2.1. Panmacular SDM Treatment

Panmacular SDM (low-intensity/high-density subthreshold diode micropulse laser) treatment employs the same treatment field (panmacular, including all retina between the major vascular arcades, including the fovea) and identical laser settings in all eyes of all patients. SDM is, therefore, a uniform treatment in all respects in each eye of every patient for every treatment indication [2,3,4,5,6,7,8,9,10,11,12,13,14,15,16]. No eye drops are required for treatment, as it is performed non-contact and without dilation [3]. By using illumination provided by the laser aiming beam, the fundus is visualized with a 90-diopter indirect lens at the laser slit lamp, and the optic nerve is identified for orientation. Then, 400–450 laser spot applications, with a 300 μm aerial diameter, an 810 nm wavelength, a power of 1.73 Watts, a 5% duty cycle, and a 0.30 s spot duration, are manually scanned or “painted” confluently in 3 passes over the panmacular retina to assure complete treatment coverage [3].

### 2.2. Standardized Automated Perimetry Testing

All included patients were experienced SAP test-takers prior to the SAP tests reported in the current study. SAP was performed using either a Humphrey-style visual field analyzer (Micromedical Devices, Calabasas, CA, USA) (used prior to 2022) or an Octopus 600 (Haag-Streit, Mason, OH, USA) (from 2022 onward), both utilizing the Swedish Interactive Threshold Algorithm (SITA Fast). All patients were tested before and after SDM treatment using the same SAP testing device and SAP program (10-2, testing the central 10 degrees of the visual field; or 24-2, testing the central 24 degrees) in the same examination room by the same technician [11]. All testing was performed with refractive correction and without pupillary dilation. In the primary clinic location, patients were treated sequentially, one eye at a time, generally treating the eye with the worst VF loss first. They then returned 1–2 weeks later for a repeat SAP to assess the treatment response in the first eye prior. If indicated, treatment of the fellow eye was performed at that time. These constitute the “unilaterally treated” group, with the untreated fellow eye serving as a control for this analysis. Bilateral simultaneous treatment was generally performed in the satellite offices visited less frequently, and for patients traveling long distances to the clinic, to expedite completion of treatment and reduce their travel burden. These eyes make up the “bilaterally treated” group. The decision to treat the subject’s eyes sequentially or simultaneously was thus determined by logistical factors rather than ocular pathology.

### 2.3. Statistical Analysis

Three groups of eyes were analyzed: all treated eyes, unilaterally treated eyes, and untreated fellow eyes of the unilaterally treated eyes (controls). Only eyes treated unilaterally on the date of the first SDM treatment were used for statistical comparison with the untreated fellow control eyes. For each group, the mean change in the difference between each study metric (MD, PSD, and MS) was calculated, and the *p*-value was derived using the *t*-test. Due to the small group size (<30 subjects each), the Mann–Whitney test was used to compare the results of the unilaterally treated and untreated control eyes. The statistical analysis was performed using R statistical software (https://www.r-project.org/ accessed on 22 January 2024). A paired Student’s *t*-test was used to determine the statistical significance of change in the pre- and post-treatment metrics in the ‘all treated eyes’, ‘unilateral treated eyes’, and ‘untreated control eyes’ treatment groups. To find the significance of the effect of the treatment between the case (unilaterally treated eyes) and control (untreated fellow eyes) groups, both a Wilcoxon rank-sum test and an unpaired *t*-test were applied to the overall difference in the case and control metrics.

## 3. Results

A total of 305 patients were identified as having undergone SAP for various indications in the study period. Using the electronic medical record to filter for eyes meeting study inclusion and exclusion criteria, 65 eyes from 34 patients were identified. Four patients (eight eyes) were excluded for poor quality/unreliable SAP testing results, and one patient (two eyes) was excluded due to non-matching pre- and post-SDM SAP testing programs (i.e., 10-2 pre- and 24-2 post-treatment). This resulted in 55 eyes of 29 patients for analysis (Table 1, Table 2 and Table 3). Patient demographic data are shown in Table 1. The pre- and post-SDM SAP metrics are shown in Table 2. Table 3 shows the results of the statistical analyses.

As shown in Figure 1, Figure 2 and Figure 3 and Table 3, nearly all treated eyes improved following SDM, with most improving by more than 2 decibels (dB). Overall, 32/36 (89%) treated eyes improved following SDM, with 23/36 (64%) improving by 2 dB or more. Of the seven treated non-OAG eyes, six out of seven (86%) improved, and five out of seven (71%) improved by >2 dB. The MD and MS improvements in treated eyes were highly significant (Table 3) (Figure 1, Figure 2, Figure 4, Figure 5 and Figure 6). Untreated fellow eye controls of the unilaterally treated eyes were significantly improved as well, but to a lesser magnitude (Table 3) (Figure 3). Comparing the unilaterally treated eyes with the untreated fellow eye controls, the treated eyes showed significant improvement (*p* = 0.016 MD and MS). PSDs were not significantly changed in any group.

## 4. Discussion

Chronic progressive diseases are the hallmark of aging [1,3]. While symptoms may occasionally wax and wane, progressive worsening over time—thus degeneration—is the rule. The CPRs that constitute the main causes of irreversible visual loss at all ages worldwide are no exception [1].

As diseases of the retina, the CPRs are all chronic neurodegenerations [2,3,4]. The macula is the greatest contributor to the NFL [5,6,7,8,9,10,11,12,13,14,15,16,17,18]. Thus, as an integrated system, it is not surprising that improvements in macular function, such as those elicited by panmacular SDM, result in improvements in the optic nerve as well.

We report the first series of eyes with visual field loss due to OAG and OA that improved following treatment. In a previous study of OAG, SDM significantly improved other quasi-objective measures such as chart visual acuity, mesopic visual acuity, and mesopic visual fields, as well as objective testing, including PERG and VER [8]. More recently, SDM VPT has been shown to reduce and even reverse loss of retinal NFL and GCC layers over time in eyes with OAG [10]. Thus, while the findings of VF improvement in OAG and OA following retinal laser treatment are novel, it would be surprising if this were not the case in the context of the other previously reported functional and anatomic improvements elicited by SDM in OAG.

The mean deviation (MD) is generally considered the most meaningful digital metric generated by SAP. This is complemented by other indices, including the mean sensitivity (MS) and pattern standard deviation (PSD) [11,12,13,14,15,16,17,18]. For the MD, the brighter the stimulus required to elicit a response, the more negative the MD values become. MD values may have a broad range of reliability from +2 dB to −30 dB. Loss of 2 dB in MD may be indicative of glaucoma in suspected cases, while annual losses of less than 1 dB per year can be indicative of disease progression [11,12,13,14,15,16,17,18]. Thus, it is notable that most SDM-treated eyes showed MD improvements of more than 2 bB (Table 2 and Table 3) (Figure 1, Figure 2, Figure 3, Figure 4, Figure 5 and Figure 6). The MS represents the average of all threshold sensitivity values across the visual field. The higher the value, the better [11]. The MS was also significantly improved in treated eyes, and like the MD, significantly improved, but less so, in untreated fellow eyes as well (Table 2 and Table 3) (Figure 1, Figure 2 and Figure 3). Visual field loss in glaucoma may be localized or heterogeneous [11]. The PSD provides a digital indicator of heterogeneity by summing the absolute value of the difference between the threshold value for each point and the average visual field sensitivity at each point (or the normal value for each point + the MD). Thus, there are two types of visual fields that will have PSDs of 0: these are normal age-adjusted fields and fields that are homogeneously depressed. Focal severe defects will generate the largest PSD [11]. In the current study, SDM treatment improved the MD and MS but not PSD. This may be due to the generalized improvements in sensitivity occurring following SDM, “lifting all ships” (Table 2 and Table 3) (Figure 4, Figure 5 and Figure 6).

Factors that might confound our findings are a placebo effect and a practice artifact [11,12,13,14,15,16,17,18]. One cannot rule out some element of the placebo effect in any test. However, while a placebo may contribute, the magnitude, frequency, and consistency of the improvements we report suggest a robust treatment effect. Because a placebo effect would be expected to predominate in the treated eye, it would not seem to explain the significant improvements in the fellow untreated eyes in this study. Nor can a practice artifact be entirely ruled out. However, the glaucoma study patients were diagnosed and managed with IOP control by their ophthalmologist, generally for years prior to inclusion in this study, suggesting that significant additional SAP learning at the time of testing in the current study was less probable. Moreover, a practice effect would be expected to manifest in both eyes more or less equally. In this light, the significant differences in pre- and post-treatment responses between treated and fellow eyes suggest that learning did not likely contribute significantly to the observed responses. The brief interval between the pre- and post-treatment SAP tests (all less than one month, with most separated by just one week) makes SDM treatment the singular ocular event to occur within the interval, minimizing other possible confounding influences such as surgery, medication, IOP changes, aging, cognitive change, or disease progression. Finally, while eyes with worse visual function tended to have more testing errors, we note that the indices of testing reliability were good overall (Table 2) (Figure 4, Figure 5 and Figure 6).

As noted above, the finding that all measures of optic nerve and ganglion cell function and anatomy, and all indices of visual function, improve in OAG following SDM, which selectively targets the panmacular retinal pigment epithelium (RPE) to normalize macular function (the “reset” effect). This indicates the presence of a modifiable retinopathy in OAG (the “retinopathy of OAG”, or ROAG) [2,3,4]. ROAG is identifiable by a reversible hyponeurotropism responsive to SDM-laser thermohormesis initiated by activation of RPE heat-shock proteins [2,3,4]. The effects of the reset phenomenon are manifold [2,3,4,8,9,10,19,20,21,22,23,24,25,26,27]. These include triggering of an acute inflammatory response (in the absence of tissue damage) to reduce chronic inflammation—a key driver of OAG, and improved RPE proteostasis and inhibition of RPE apoptosis via up-regulation of the endoplasmic reticulum unfolded protein response, to improve all measures of cell function including mitochondrial function [3]. In laboratory studies, SDM elicits both local and systemic therapeutic immunomodulation [19,20,21]. Clinically, and in every reported human, animal, and tissue study of pulsed microsecond laser sublethal to the RPE (such as SDM), all measured variables improve toward normal, healthy levels [2,3,4,8,9,10,19,20,21,23,24,25,26]. Thus, adverse treatment effects have not been observed following SDM [2,3]. By improving RPE function, critical RPE paracrine functions essential for normal ocular function, such as cytokine expression and response, are also normalized. This suggests SDM may reverse an intrinsic hyponeurotropism of retinal origin in OAG that may cause glaucomatous optic atrophy to progress despite IOP lowering [3,8,9,10].

Improvements in untreated fellow eyes by all measures have been previously identified following SDM, including by electrophysiology, following treatment of just one eye in dry AMD, IRDs, and OAG [2,3,4]. Of particular interest in this regard is an animal study by Caballero and associates in which therapeutic systemic immunoactivation, manifested by the recruitment of bone marrow-derived immune cells to both eyes, was found to occur following SDM treatment of just one eye [21]. This type of systemic therapeutic immunoactivation could explain the significant, but less robust, improvements in the VF of untreated fellow control eyes in the current study. Finally, the optic nerve contains efferent as well as afferent fibers, and this communication can influence visual function in the fellow eye [28,29]. The rapid response of the fellow eye to SDM in the contralateral eye (often within one week) may favor the possibility of such efferent interocular communication.

While SDM may improve the function of living, dysfunctional cells, the resurrection of dead cells is unlikely [3,8,9,10]. Therefore, the rapid (often within one week) improvements in visual function observed in the current study following a single treatment of SDM suggest restored function to moribund, possibly pre-apoptotic ganglion cells and their axons that are sufficiently dysfunctional to not register visual function on SAP prior to SDM [3,5,6,8,9,10]. Such functional restoration would represent “neuroenhancement” [5,6] (Figure 4, Figure 5 and Figure 6). The maintenance of neuroenhancement is expected to achieve neuroprotection or resistance to progressive optic atrophy over time, thereby reducing the risks of vision loss [5,6]. The goal of clinical therapy in OAG would, therefore, be to produce repeated temporary reversals of disease progression, such as those produced by SDM, reflected in the SAP improvements reported here, that are additive over time to result in slowing of disease progression [3,8,9,10]. As noted previously, this hypothesis is supported by a recent report describing positive NFL and GCC OCT thickness trends in eyes with OAG undergoing long-term SDM [10] (Figure 4). In non-OAG OA, rather than prevention of progression, the clinical goal is to produce temporary, symptomatically valuable, and ideally renewable improvements in visual function (Figure 5 and Figure 6).

This report suffers from weaknesses inherent to retrospective studies, including small sample sizes, potential for case selection bias, short study duration, and the absence of a prospective study protocol. However, the results are robust and consistent with prior studies [3,8,9,10]. The improvements in OAG and OA we report following SDM suggest safe and effective neuroenhancement and the potential for robust therapeutic neuroprotection—a long-sought-after goal in OAG [5,6,7,8,9,10]. We hope our findings will inspire further prospective studies needed to confirm or refute these findings and determine if the implementation of effective neuroprotection can indeed reduce visual loss from OAG better than IOP control alone.

## Figures and Tables

**Figure 1 diagnostics-15-00912-f001:**
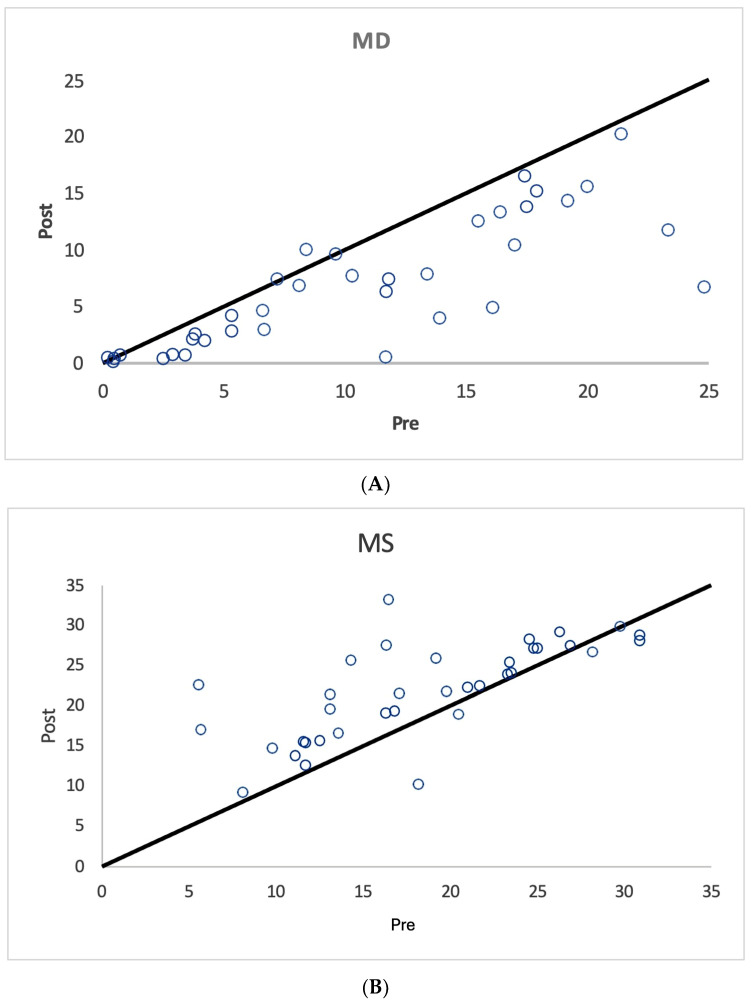
Scatter graph of mean deviation (MD) (**A**) and mean sensitivity (MS) (**B**) values in decibels before and after SDM treatment, all treated eyes. Note significant improvements in MD and MS values after treatment.

**Figure 2 diagnostics-15-00912-f002:**
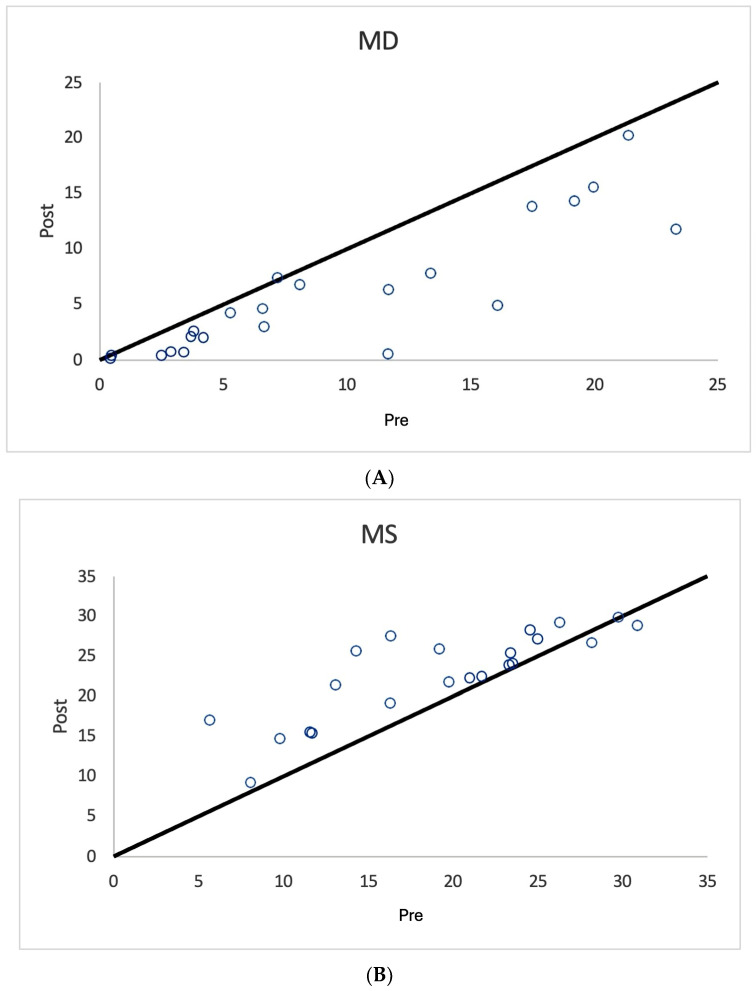
Scatter graph of mean deviation (MD) (**A**) and mean sensitivity (MS) (**B**) values in decibels before and after SDM treatment, unilaterally treated eyes. Note significant improvements in MD and MS values after treatment.

**Figure 3 diagnostics-15-00912-f003:**
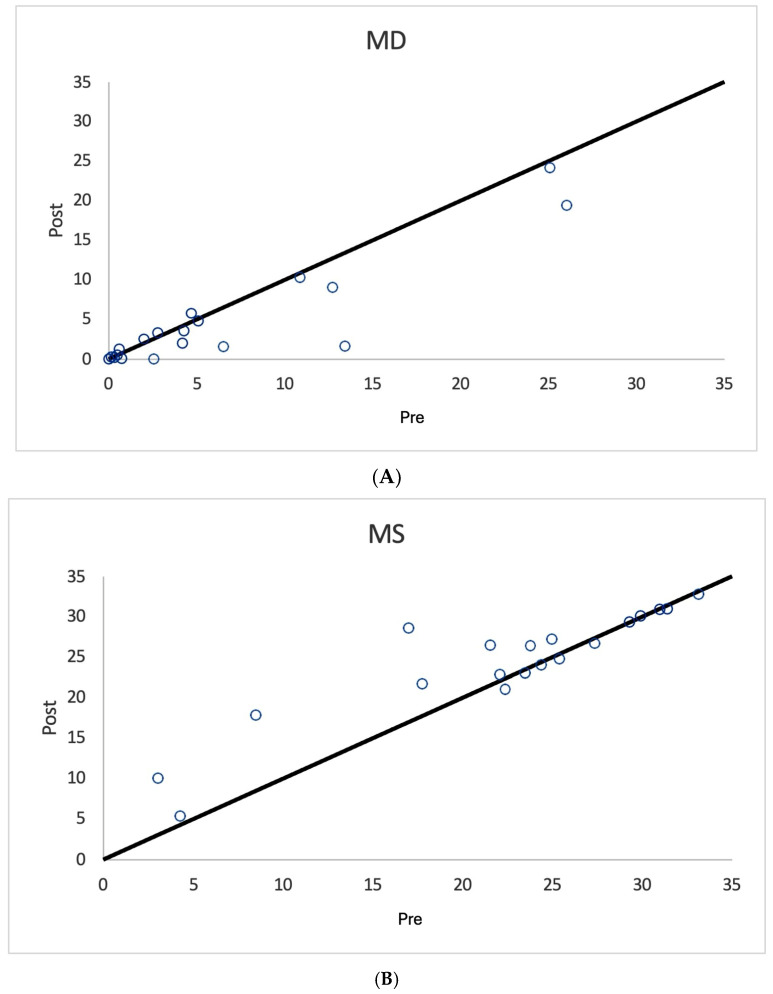
Scatter graph of mean deviation (MD) (**A**) and mean sensitivity (MS) (**B**) values in decibels before and after SDM treatment, untreated fellow eye controls. Note significant improvements in MD and MS values after treatment, but less significant than treated eyes.

**Figure 4 diagnostics-15-00912-f004:**
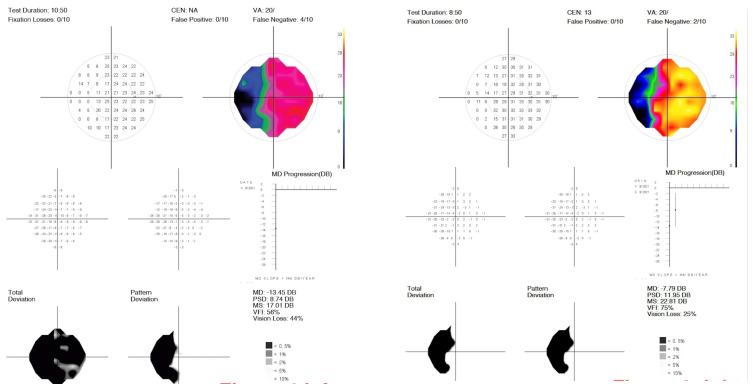
A 73-year-old male with mild non-proliferative diabetic retinopathy without macular edema, early age-related macular degeneration, prior cerebral vascular occlusion with left homonymous hemianopsia, and open-angle glaucoma with glaucomatous optic neuropathy. 10-2 Humphrey-style SAP with false color mapping, left side before and right side 1 week after panmacular SDM. Note improvement in SAP following treatment.

**Figure 5 diagnostics-15-00912-f005:**
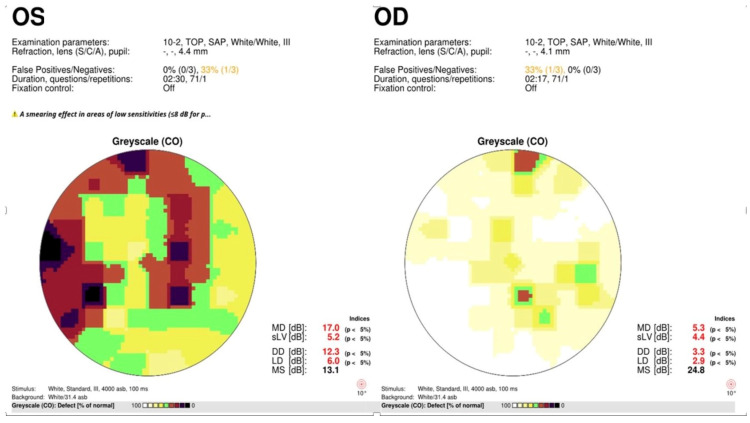
A 52-year-old female with visual disability due to multiple sclerosis. Top row, Octopus 600 10-2 SAP of right and left eye before same-day panmacular SDM OU. Bottom row, 17 days after SDM. Patient reported symptomatic improvement in both eyes, noted the day after treatment. Note VF improvement by SAP in both eyes following treatment.

**Figure 6 diagnostics-15-00912-f006:**
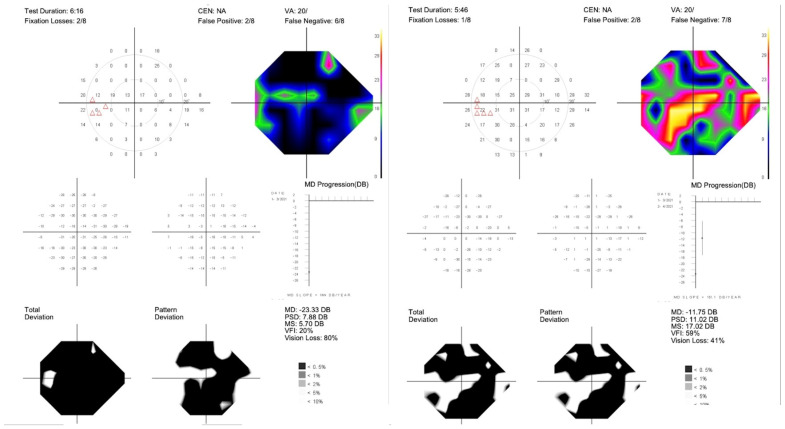
A 74-year-old male with a history of optic atrophy from non-arteritic idiopathic anterior ischemic optic neuritis in both eyes for 8 years. Visual acuities were finger counting in his right eye and 20/70 in his left eye. Panmacular SDM was performed in the left eye in the hope of improving visual function. 10-2 Humphrey-style SAP VFs with false color mapping before (left side) and 1 month after (right side) SDM treatment. Note VF improvement by SAP following treatment. Visual acuity was also improved to 20/60 in the left eye. SDM was continued as vision protection therapy every 3–4 months, during which time the visual acuity continued to gradually improve. A total of 14 months after initial treatment, the visual acuity had improved to 20/30 and was maintained until the most recent visit, 3 years after initial SDM treatment. From reference [3].

**Table 1 diagnostics-15-00912-t001:** Patient demographics.

No. Patients	29
Caucasian	24 (83%)
Hispanic	5 (17%)
Total Study Eyes	55
Total Treated Eyes	36
Bilaterally SDM-treated Eyes	14/36 (39%)
Unilaterally SDM-treated eyes	22/36 (61%)
Untreated Control Eyes	19
Female	17 (59%)
Age (years)	62–89 (avg 76, SD 7.5)
Total OD	26 (47%)
Total OS	29 (53%)
Total Treated Eyes OAG	29 eyes (81%)
Total Treated Eyes Other OA	7 eyes (19%) (AION 3 pts, 5 eyes, Mult Scl 1 pt 2)
Total Pseudophakic Eyes	25 eyes (46%)
10-2	26 eyes (47%)
24-2	29 eyes (53%)

No. = number; OD = right eye; OS = left eye; OAG = open-angle glaucoma; OA = non-glaucomatous optic atrophy; AION = optic atrophy post non-arteritic ischemic optic neuropathy; Mult Scl = optic atrophy associated with multiple sclerosis; SDM = panmacular low-intensity/high-density subthreshold diode micropulse laser; 10-2, automated perimetric testing of the central 10 degrees of the visual field; 24-2, automated perimetric testing of the central 24 degrees of the visual field; pt = patient.

**Table 2 diagnostics-15-00912-t002:** Standardized Automated Perimetry Before and After Panmacular SDM for Neuroprotection in Open-Angle Glaucoma and Other Optic Atrophies.

		Pre MD	Pre PSD	Pre MS	False ±	Post MD	Post PSD	Post MS	False ± (%)
All Treated Eyes *n* = 36	Mean Median SD	10.5 9.5 7.0	6.0 6.4 2.8	18.4 17.7 7.0	9.2 0 15.5	6.9 6.5 5.5	5.7 5.3 3.1	21.9 22.4 5.9	9.9 0 14.8
Unilaterally Treated Eyes *n* = 22	Mean Median SD	9.5 6.9 7.1	5.4 5.4 2.6	19.3 20.4 7.0	8.5 0 16.0	5.9 4.4 5.7	4.9 4.15 2.7	22.8 24 5.4	10 0 15.6
Fellow Eye Controls *n* = 19	Mean Median SD	6.46 4.2 7.7	4.2 3.4 3.4	22.2 23.8 8.5	3.7 0 11.7	4.7 2.0 6.5	3.8 3.18 2.9	24.2 26.4 6.8	6.0 0 14.1

MD = mean deviation in decibels; PSD = pattern standard deviation in decibels; MS = mean sensitivity decibels; *n* = number. False ± = percentage of false positive and/or negative responses from total responses detected by automated perimetric testing.

**Table 3 diagnostics-15-00912-t003:** Results of statistical analysis.

Study Groups		SAP Metrics		
All Treated Eyes		MD	PSD	MS
	Mean difference	3.651944	0.2786111	−3.48528
	*p*-value	7.02 × 10^−6^	0.44	7.02 × 10^−6^
	95% CI	[2.25, 5.06]	[−0.45, 1.01]	[−5.21, −1.76]
Unilaterally Treated Eyes				
	Mean Difference	3.612273	0.4277273	−3.53909
	*p*-value	1.00 × 10^−4^	0.28	3.95 × 10^−4^
	95% CI	[2.04, 5.18]	[−0.37, 1.22]	[−5.29, −1.79]
Unilateral Untreated Fellow Eye Controls				
	Mean Difference	1.724211	0.4215789	−2.04368
	*p*-value	0.03	0.23	0.03
	95% CI	[0.19, 3.26]	[−0.29, 1.14]	[−3.82, −0.27]

SAP = standardized automated perimetry; MD = mean deviation in decibels; PSD = pattern standard deviation in decibels; MS = mean sensitivity decibels; CI = confidence interval.

## Data Availability

The raw study data are available as Appendix A submitted with the study manuscript.

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
