# Peer review of "Visual Field Improvement by Standardized Automated Perimetry Following Panmacular Subthreshold Diode Micropulse Laser (SDM) in Open-Angle Glaucoma and Other Optic Atrophiesâ€"

_diagnostics, 2025, doi:10.3390/diagnostics15070912_

Round 1
Reviewer 1 Report
Comments and Suggestions for Authors
- Overall this is an interesting and novel study evaluating SDM in improving visual field in glaucoma primarily and also optic atrophies in a group of 28 patients (54 eyes). This is quite novel and has not been evaluated previously and contributes significantly to the literature of this field. I applaud the authors for this work.
- Visual field testing can vary significantly even without treatment. It would be best to include in the study multiple measurements for baseline and post intervention to ensure that the change noted is not just testing variation. Having a single test before/after SDM make the results less compelling.
- With visual field testing, many studies often require individual to take it a couple times first as patients get used to visual field testing and there can be a significant learning curve. Please provide information on how many visual fields patients had prior to the baseline visual field testing. The authors simply state “All included patients were experienced SAP test-takers prior to the SAP tests reported in the current study.” Would a single SAP prior be considered experienced? What was the mean number of SAP tests taken? If none or this information is not available, this improvement could be driven primarily by the learning curve as patients learn how to perform visual field testing.
- Please add in the introduction and conclusions additional references and theory behind how SDM improves function of the ganglion cell, nerve fiber layer, and inner retinal functioning. Retinal pigment epithelial absorption in RPE diseases makes sense from a mechanistic standpoint for SDM treatment, but the theory behind it for glaucoma appears less obvious and is not clearly addressed in the manuscript.
- It is unclear from the manuscript for what etiology the SDM was being applied in these patients. Please explicitly state this. Was the SDM being used to treat the OAG or OA? Or did all these patients also have a concurrent retinal disease that was being treated with SDM and also happened to have OAG or OA by happenstance? It appears that most of the concurrent retinal diseases were excluded, so I would assume the treatment was specifically performed for OAG or OA, but please state explicitly.
- Please provide a patient flow diagram indicating how many patients initially met the inclusion criteria and how many patients were excluded for each of the exclusion criteria (Exclusions were VA of 20/200 or worse, poor-quality / unreliable testing, prior SDM or other macular laser treatment, prior or current intraocular anti-vascular endothelial growth factor (VEGF) or steroid injection, and/or other obfuscating ocular disease such as significant cataract, or macular disease such retinal vascular occlusion, epiretinal membrane, macular edema, significant diabetic retinopathy, prior retinal detachment, or advanced AMD) so that readers can better understand possible selection bias.
- It seems atypical that the “the decision whether to treat and test one eye at time, or both eyes simultaneously, was based on office location, not ocular pathology.” Why would office location have dictated treating both vs a single eye? Given this difference in office location, it would be helpful in the methods to provide a sentence or two about the office locations and any differences between the patient groups who go to each one for possible biases that could exist based on this difference.
- In Table 1, please update it to present the information better. For example, on many of the items like female, give both the raw number along with the percentage. Given the number of 28 patients/54 eyes, it is easier for readers to follow percentages of patients/eyes rather than raw numbers. On age, please include the standard deviation along with the mean and range. If it is available, please give information on race/ethnicity of the patients.
- The abbreviation MS is used to mean 2 different things in the manuscript. In Table 1 it indicates multiple sclerosis but in the rest of the manuscript it represents mean sensitivity. This will be confusing to readers. I would recommend removing or changing 1 of the MS abbreviations.
- In Table 1, for the 2 patients with optic atrophy due to multiple sclerosis, did these patients have optic neuritis associated with multiple sclerosis? If so, it might be better to categorize these as “optic atrophy due to optic neuritis associated with multiple sclerosis”.
- In Table 1, rather than putting IOL with 25 patients, I would recommend changing it to be pseudophakic.
- Can the authors explain why the controls (unilateral untreated fellow eyes) improved after treatment (Figure 3)? This is concerning since this could indicate that patients are learning the visual field testing and thus their testing in general is improving, not indicating a treatment effect but rather a testing improvement.
- Figure 4 shows a patient with mild NPDR and AMD, but from the exclusion criteria shouldn’t this patient have been excluded since macular disease, advanced diabetic retinopathy, and advanced AMD are all exclusion criteria. Please specify in more detail what is meant by advanced for both diabetic retinopathy and AMD. What stage diabetic retinopathy? Advanced AMD was neovascular AMD or GA?
- To ensure patient anonymity, please delete anything that could possibly be used to identify patients. Please delete from Figure 4 and Figure 6 from the visual fields: date of birth, date of exam, and time of exam. From Figure 5, please delete the date and time of the exams. If you want that information, please include more general information like patient age and the year/month of exam.
- I would recommend deleting the red lettering on Figures 4, 5, and 6 where Figure 4 states “Figure 4 left” and “Figure 4 right” (and figure 6 states the same) and Figure 5 states “Figure 5 top” and bottom. This can be deleted as you have a/b to label them.
- On the Figure 6 legend, the authors state “non-arteritic idiopathic anterior ischemic optic neuritis”. Most commonly this is referred to as “non-arteritic idiopathic anterior ischemic optic neuropathy” rather than neuritis. Is there a reason the authors are not using this conventional diagnosis?
- In addition to visual field testing results, it would be helpful to include other results of visual function, such as visual acuity before/after SDM to better bolster the case that this improves vision. The authors briefly mention it in Figure 6 legend on that one patient, but do the authors have mean visual acuity before/after or other measures of visual function additionally? If you have it, this would be helpful to include.
- Why did the authors select the short term (primarily 1 week, some up to 1 month) of follow-up for this study? It would be helpful also to consider how durable the treatment is and how long these effects last, so if longitudinal monitoring was performed or is available, this would also be helpful to consider.
- Please further elaborate in the weaknesses paragraph (last paragraph beginning with “This report suffers from weaknesses inherent to retrospective studies…”) about further limitations in the study. Some ones that could be added include the lack of longitudinal follow-up, selection bias, and a single visual field testing.
Author Response
Diagnostics-3533436
Point-by-point responses to Review #1 comments:
We thank the Editor for the opportunity to address the reviewer questions. Point-by-point responses to each reviewer are provided below.
Comments and Suggestions for Authors
- Overall this is an interesting and novel study evaluating SDM in improving visual field in glaucoma primarily and also optic atrophies in a group of 28 patients (54 eyes). This is quite novel and has not been evaluated previously and contributes significantly to the literature of this field. I applaud the authors for this work.
Answer 1: We thank the reviewer for his comment.
- Visual field testing can vary significantly even without treatment. It would be best to include in the study multiple measurements for baseline and post intervention to ensure that the change noted is not just testing variation. Having a single test before/after SDM make the results less compelling.
Answer 2: We completely agree with the reviewer and acknowledge this deficiency in the paper. As stated in the paper (Discussion lines 234-248), this is a retrospective study of patients managed in clinical practice who were not / could not be asked to undergo repeated baseline testing for research purposes. We note the various sources of error and discuss how these might affect the results.
- With visual field testing, many studies often require individual to take it a couple times first as patients get used to visual field testing and there can be a significant learning curve. Please provide information on how many visual fields patients had prior to the baseline visual field testing. The authors simply state “All included patients were experienced SAP test-takers prior to the SAP tests reported in the current study.” Would a single SAP prior be considered experienced? What was the mean number of SAP tests taken? If none or this information is not available, this improvement could be driven primarily by the learning curve as patients learn how to perform visual field testing.
Answer 3: We agree that repeated SAP testing would be ideal. Ideal can only be approached in prospective study, while this is a retrospective study. As a study of the results in clinical practice such repeated testing is not possible for a number of reasons. The number and frequency of prior SAP tests per patient could not be determined as the vast majority of prior examinations were performed elsewhere in other practices over many years. We acknowledge this limitation in the discussion. To further clarify and address this concern, the additional text has been added to the Discussion, lines 235-243: “However, while placebo may contribute, the magnitude, frequency, and consistency of the improvements we report suggest a robust treatment effect. Neither can practice artifact be entirely ruled out. However, all glaucoma study patients were diagnosed and managed with IOP control by their ophthalmologist generally for years prior to inclusion in this study, suggesting that significant additional SAP learning at the time of testing in current study was less likely. Further, the significant difference in pre and post treatment responses between treated and fellow eyes and the magnitude and consistency of the treatment responses suggest that learning did not contribute significantly to the observed treatment responses..”
- Please add in the introduction and conclusions additional references and theory behind how SDM improves function of the ganglion cell, nerve fiber layer, and inner retinal functioning. Retinal pigment epithelial absorption in RPE diseases makes sense from a mechanistic standpoint for SDM treatment, but the theory behind it for glaucoma appears less obvious and is not clearly addressed in the manuscript.
Answer 4: We agree with the reviewer that the mechanism of action of SDM in OAG (and other disorders) is very interesting. Thus, references 2, 3, 8-10 each include extensive analyses and discussions of the theory and mechanism of action of SDM in OAG regarding NFL, GCC, and inner retinal functioning. These references include entire books and book chapters devoted to the topic, as well as peer-reviewed published studies. This is also discussed extensively in the Introduction lines 55-69 and Discussion lines 243-271.
- It is unclear from the manuscript for what etiology the SDM was being applied in these patients. Please explicitly state this. Was the SDM being used to treat the OAG or OA? Or did all these patients also have a concurrent retinal disease that was being treated with SDM and also happened to have OAG or OA by happenstance? It appears that most of the concurrent retinal diseases were excluded, so I would assume the treatment was specifically performed for OAG or OA, but please state explicitly.
Answer 5: As stated in the title of the report, we report SDM for visual field loss both in eyes with OAG and OA. Thus, while minor concurrent conditions may have been present the primary indication for treatment was visual field loss from OAG or OA. The rationale for SDM treatment is stated in the Methods, lines 94-97: “The indication for SDM treatment in the eyes examined in this study was to slow disease progression, predicted by prior studies showing improvements in prognostic surrogates of disease worsening following SDM, including all measures of retinal, optic nerve, retinal ganglion cell, and visual function in multiple disorders including OAG. [2-10] “
That treatment was specifically performed for OAG and OA is stated in line 99:“Inclusion criteria were a prior diagnosis of OAG or OA…”
The inclusion and exclusion criteria address possible other pathologic associations is described in lines 99-108.
- Please provide a patient flow diagram indicating how many patients initially met the inclusion criteria and how many patients were excluded for each of the exclusion criteria (Exclusions were VA of 20/200 or worse, poor-quality / unreliable testing, prior SDM or other macular laser treatment, prior or current intraocular anti-vascular endothelial growth factor (VEGF) or steroid injection, and/or other obfuscating ocular disease such as significant cataract, or macular disease such retinal vascular occlusion, epiretinal membrane, macular edema, significant diabetic retinopathy, prior retinal detachment, or advanced AMD) so that readers can better understand possible selection bias.
The following has been added to the Results section lines 159-162, which we believe more succinctly addresses the question of selection bias: “305 patients were identified as having undergone SAP for various indications in the study period. 64 eyes of 33 patients met criteria for study inclusion. 4 patients (8 eyes) were excluded for poor quality / unreliable SAP testing results and 1 patient (2 eyes) was excluded due to non-matching pre and post SDM SAP testing programs (ie, 10-2 pre and 24-2 post treatment), resulting in 54 eyes of 28 for analysis (Tables 1-3). “
- It seems atypical that the “the decision whether to treat and test one eye at time, or both eyes simultaneously, was based on office location, not ocular pathology.” Why would office location have dictated treating both vs a single eye? Given this difference in office location, it would be helpful in the methods to provide a sentence or two about the office locations and any differences between the patient groups who go to each one for possible biases that could exist based on this difference.
Answer 7: As stated in the Methods, the decision to treat one or both eyes in a single visit was dictated by the logistics of clinical practice rather than ocular pathology. Additional clarification as been added to the Methods lines 133-140: “The decision to treat each eye sequentially or simultaneously was dictated by logistics rather than ocular pathology. Sequential treatment of one eye at a time was preferred, with post treatment SAP performed after treatment of the first eye, generally the eye with the greatest pretreatment visual field loss. These constitute the “unilaterally treated” group, with the untreated fellow eye serving as a control for this analysis. Bilateral simultaneous treatment was generally performed in the satellite offices visited less frequently, and for patients travelling long distances to the clinic, to expedite treatment completion and reduce their travel burden. These eyes make up the “bilaterally treated” group.”
- In Table 1, please update it to present the information better. For example, on many of the items like female, give both the raw number along with the percentage. Given the number of 28 patients/54 eyes, it is easier for readers to follow percentages of patients/eyes rather than raw numbers. On age, please include the standard deviation along with the mean and range. If it is available, please give information on race/ethnicity of the patients.
Answer 8: Done
- The abbreviation MS is used to mean 2 different things in the manuscript. In Table 1 it indicates multiple sclerosis but in the rest of the manuscript it represents mean sensitivity. This will be confusing to readers. I would recommend removing or changing 1 of the MS abbreviations.
Answer 9: MS has been changed so as not to refer to multiple sclerosis.
- In Table 1, for the 2 patients with optic atrophy due to multiple sclerosis, did these patients have optic neuritis associated with multiple sclerosis? If so, it might be better to categorize these as “optic atrophy due to optic neuritis associated with multiple sclerosis”.
Answer 10: These patients had longstanding vision and visual field loss and optic atrophy due to multiple sclerosis, as stated in the text. Information about the specific cause of the MS-related optic atrophy was not available, but was assumed to be past retrobulbar optic neuritis.
- In Table 1, rather than putting IOL with 25 patients, I would recommend changing it to be pseudophakic.
Answer 11. Done
- Can the authors explain why the controls (unilateral untreated fellow eyes) improved after treatment (Figure 3)? This is concerning since this could indicate that patients are learning the visual field testing and thus their testing in general is improving, not indicating a treatment effect but rather a testing improvement.
Answer 12. This is a very interesting question not only regarding possible placebo / learning effect, but from a biologic perspective. We discuss this concern in the Discussion. Lines 237-243 discuss the possibility of confounding influences, noting both that the patients in this study were experienced SAP test takers reducing the potential for a practice effect, and that the significant difference in the results between treated vs control eyes indicates that learning / practice artifact, if present, did not significantly affect the study results. Lines 250-278 discuss the biologic implications of this finding.
- Figure 4 shows a patient with mild NPDR and AMD, but from the exclusion criteria shouldn’t this patient have been excluded since macular disease, advanced diabetic retinopathy, and advanced AMD are all exclusion criteria. Please specify in more detail what is meant by advanced for both diabetic retinopathy and AMD. What stage diabetic retinopathy? Advanced AMD was neovascular AMD or GA?
Answer 13. Lines 103-108 of the Methods note that eyes with “macular edema,…significant diabetic retinopathy, and advanced AMD” were excluded. Thus, this patient met inclusion criteria as mild NPDR and early AMD would not be expected to significantly affect SAP results. Advanced AMD includes GA and neovascular AMD and were excluded at these would generally influence SAP results. The diabetic retinopathy exclusion has been clarified with the addition of “significant diabetic retinopathy (beyond “early” non-proliferative stage),..”
- To ensure patient anonymity, please delete anything that could possibly be used to identify patients. Please delete from Figure 4 and Figure 6 from the visual fields: date of birth, date of exam, and time of exam. From Figure 5, please delete the date and time of the exams. If you want that information, please include more general information like patient age and the year/month of exam.
Answer 14: Done
- I would recommend deleting the red lettering on Figures 4, 5, and 6 where Figure 4 states “Figure 4 left” and “Figure 4 right” (and figure 6 states the same) and Figure 5 states “Figure 5 top” and bottom. This can be deleted as you have a/b to label them.
Answer 15: Done
- On the Figure 6 legend, the authors state “non-arteritic idiopathic anterior ischemic optic neuritis”. Most commonly this is referred to as “non-arteritic idiopathic anterior ischemic optic neuropathy” rather than neuritis. Is there a reason the authors are not using this conventional diagnosis?
Answer 16: We believe the difference is semantic. We refer to the proximate cause of the subsequent optic atrophy, which was optic neuritis (“optic atrophy…from… optic neuritis”). Neuropathy refers to the syndrome in general.
- In addition to visual field testing results, it would be helpful to include other results of visual function, such as visual acuity before/after SDM to better bolster the case that this improves vision. The authors briefly mention it in Figure 6 legend on that one patient, but do the authors have mean visual acuity before/after or other measures of visual function additionally? If you have it, this would be helpful to include.
Answer 17: Data on visual acuity, mesopic visual acuity, mesopic visual fields, pattern electroretinography, visually evoked potential, and long-term OCT nerve fiber layer and ganglion cell complex layer trends have already been reported and all shown to significantly improve following SDM in OAG (refs 3,8,9,10). This paper concerns only SAP effects.
- Why did the authors select the short term (primarily 1 week, some up to 1 month) of follow-up for this study? It would be helpful also to consider how durable the treatment is and how long these effects last, so if longitudinal monitoring was performed or is available, this would also be helpful to consider.
Answer 18: Decades of experience with SDM for many difference indications shows that the treatment effects can generally be measured within 24 hours of treatment. Thus, waiting beyond a week or so to assess the response by SAP is generally unnecessary. Long term serial SAP testing is not done rountinely as it is generally unnecessary and burdensome; and other essential objective and easily monitored long-term disease indicators, such retinal NFL and GCC layer thickness trends (ref 10), show that SDM causes significant long-term progressive reversal of optic nerve damage in OAG.
- Please further elaborate in the weaknesses paragraph (last paragraph beginning with “This report suffers from weaknesses inherent to retrospective studies…”) about further limitations in the study. Some ones that could be added include the lack of longitudinal follow-up, selection bias, and a single visual field testing.
Answer 19: Discussion lines 294-296 states: “This report suffers from weaknesses inherent to retrospective studies, including small sample sizes, case selection, absence of a prospective study protocol, and absence of confirmatory serial testing.” We believe this addresses each of the reviewer’s concerns directly. However, in an attempt to further clarify we have amended the text to state: “This report suffers from weaknesses inherent to retrospective studies, including small sample sizes, potential for case selection bias, and the absence of a prospective study protocol and long-term serial testing. These must be addressed with further study which we hope the current study encourages.”
Reviewer 2 Report
Comments and Suggestions for Authors
The manuscript presents a retrospective study evaluating the impact of Panmacular Subthreshold Diode Micropulse Laser (SDM) on visual fields in patients with Open Angle Glaucoma (OAG) and other optic atrophies (OA). The study is well-structured, methodologically sound, and provides meaningful insights into an area that lacks extensive research. The findings contribute to the ongoing discussion regarding neuroprotection and neuroenhancement in ophthalmic treatments. However, several areas require clarification and improvement for greater scientific rigor and readability.
- Please strengthen the significance of the study in the introduction section and clarify the advancement of this study over previous ones, given that “examples of improved VF by SAP in OAG, OA, and CPRs have been described following SDM.” A clear differentiation between prior case reports and the current systematic evaluation should be provided, emphasizing how this study uniquely contributes to the field.
- The patient number and eye number do not match in Table 1 and Table 2. In Table 1, No. Eyes (54) is not equal to the sum of treated eyes (36) and fellow eyes (19). Also, the Unilateral SDM number does not equal the fellow control eye number. The sum of Bilateral patient number (14/2=7) and unilateral patient number (22) does not match the total patient number (28). The N number in Table 2 is also inconsistent with the patient number in Table 1. Please verify all data and ensure internal consistency across tables.
- The meaning of "10-2" and "24-2" in Table 1 should be explicitly stated. Similarly, "False" in Table 2 needs clarification. Additionally, "OU" in the abstract should be defined.
- The authors discuss the potential systemic influence of SDM treatment on the contralateral eye, yet they use the fellow eye in unilateral treatment as a control. This raises concerns about whether the study design adequately isolates the treatment effect. Would a more scientifically rigorous comparison involve a separate control group of patients who received no treatment in either eye?
Author Response
Diagnostics-3533436
Point-by-point responses to Review #2 comments:
We thank the Editor for the opportunity to address the reviewer questions. Point-by-point responses to each reviewer are provided below.
The manuscript presents a retrospective study evaluating the impact of Panmacular Subthreshold Diode Micropulse Laser (SDM) on visual fields in patients with Open Angle Glaucoma (OAG) and other optic atrophies (OA). The study is well-structured, methodologically sound, and provides meaningful insights into an area that lacks extensive research. The findings contribute to the ongoing discussion regarding neuroprotection and neuroenhancement in ophthalmic treatments. However, several areas require clarification and improvement for greater scientific rigor and readability.
We thank the reviewer for his comments and suggestions, which we address below in a point-by-point fashion:
- Please strengthen the significance of the study in the introduction section and clarify the advancement of this study over previous ones, given that “examples of improved VF by SAP in OAG, OA, and CPRs have been described following SDM.” A clear differentiation between prior case reports and the current systematic evaluation should be provided, emphasizing how this study uniquely contributes to the field.
Answer 1: As requested by the reviewer, we have attempted to further clarify the reasoning behind the study in the introduction with the following, lines 75-82: “Despite these case reports, there has been no systematic study of the effects of SDM on VFs by SAP. In the current study we retrospectively examine and statistically analyze VF results by SAP in eyes with OAG and OA before and after initial treatment with SDM in an effort to better understand if SAP improvements following SDM in OAG and OAG are aberrations or represent a characteristic therapeutic response to treatment. Because visual field loss is a hallmark of disease progression in OAG the ability to therapeutically reverse visual field loss would represent an important advance in disease management and vision loss prevention in OAG. “
- The patient number and eye number do not match in Table 1 and Table 2. In Table 1, No. Eyes (54) is not equal to the sum of treated eyes (36) and fellow eyes (19). Also, the Unilateral SDM number does not equal the fellow control eye number. The sum of Bilateral patient number (14/2=7) and unilateral patient number (22) does not match the total patient number (28). The N number in Table 2 is also inconsistent with the patient number in Table 1. Please verify all data and ensure internal consistency across tables.
Answer 2: We are indebted to Review #2 for identifying this mistake, which has been corrected.
- The meaning of "10-2" and "24-2" in Table 1 should be explicitly stated. Similarly, "False" in Table 2 needs clarification. Additionally, "OU" in the abstract should be defined.
Answer 3: We thank the review for these suggestions. We have defined 10-2 and 24-2 in the legend of Table 1. “OU” in the abstract has been replaced with “in both eyes.” “False” in Table 2 has defined in the table legend as “False +/- = percentage of false positive and / or negative responses from total responses detected by automated perimetric testing. “
- The authors discuss the potential systemic influence of SDM treatment on the contralateral eye, yet they use the fellow eye in unilateral treatment as a control. This raises concerns about whether the study design adequately isolates the treatment effect. Would a more scientifically rigorous comparison involve a separate control group of patients who received no treatment in either eye?
Answer 4: While we agree that a more rigorous prospectively designed study is always preferrable, the current study is retrospective, done in hopes of providing a rationale for future prospective study, as we note in the Discussion, line 305. While the fellow eye is an imperfect control due to the potential for bilateral treatment effects, we believe some control is better than no control and provides useful, although not definitive, information. This discussion has been amplified in the text to better address the reviewer’s concern, Discussion lines 243-260: “Factors that might confound our findings are a placebo effect, and practice artifact. [11-18] One cannot rule out some element of placebo effect in any test. However, while placebo may contribute, the magnitude, frequency, and consistency of the improvements we report suggest a robust treatment effect. Because a placebo effect would be expected to manifest in the treated eye, it would not seem to explain the significant improvements in the fellow, untreated eyes. Neither can a practice artifact be entirely ruled out. However, the glaucoma study patients were diagnosed and managed with IOP control by their ophthalmologist generally for years prior to inclusion in this study, suggesting that significant additional SAP learning at the time of testing in current study was less likely. Moreover, a practice effect would be expected to manifest in both eyes more or less equally. Thus, the significant differences in pre and post treatment responses between treated and fellow eyes suggest that learning did not contribute significantly to the observed responses. The brief interval between the pre and post treatment SAP tests (all less than one month, most separated by just one week) makes SDM treatment the signal ocular event to occur within the interval, minimizing other possible confounding influences such as surgery, medication, IOP change, aging, cognitive change, or disease progression. Finally, while eyes with worse visual function tended to have more testing errors, we note the indices of testing reliability were good overall.[24] (Table 2) (Figures 4-6)”
Submission Date
Round 2
Reviewer 1 Report
Comments and Suggestions for Authors
The authors have thoroughly and comprehensively responded to all of the comments of the reviewers. The authors have performed a major revision incorporating all of the recommendations and edits into the manuscript. The revised manuscript is much improved and contributes significantly to the literature of the field. The updated manuscript addresses all of the concerns that were raised and better provides details that are requested by the reviewers. I applaud the authors for this interesting and compelling work. This contributes to the literature of the field.